# Effects of the Thickness of *N*,*N*′-diphenyl-*N*,*N*′-di(m-tolyl)-benzidine on the Electro-Optical Characteristics of Organic Light-Emitting Diodes

**DOI:** 10.3390/ma12060966

**Published:** 2019-03-22

**Authors:** Sang-Geon Park, Won Jae Lee, Min Jong Song, Johngeon Shin, Tae Wan Kim

**Affiliations:** 1Division of Smart Electrical and Electronic, Silla University, 140 Baegyang-daero 700beon-gil, Sasang-gu, Busan 46958, Korea; sgpark7@hotmail.com; 2Electrical and Computer Engineering, Stony Brook University, 100 Nicolls Rd, Stony Brook, NY 11794, USA; wonjae.lee@stonybrook.edu; 3Department of Radiation Technology, Gwangju Health College, 73 Bungmun-daero 419 beongil, Gwangsangu, Gwangju 13557, Korea; smj@ghu.ac.kr; 4Department of Materials Science and Engineering, Silla University, Busan 46958, Korea; jeshin@silla.ac.kr; 5Department of Information Display Engineering, Hongik University, Seoul 04066, Korea

**Keywords:** organic light-emitting diode, current density, current efficiency

## Abstract

We examined the electro-optical characteristics of organic light emitting diodes according to the *N*,*N*′-diphenyl-*N*,*N*′-di(m-tolyl)-benzidine (TPD) thicknesses. The thicknesses of TPD were varied from 5 nm to 50 nm. The current density of the device with a TPD thickness of 5 nm was 8.94 times higher than that with a thickness of 50 nm at a driving voltage of 10 V. According to the conduction–current characteristics of conductors, the current densities improved with a decreasing TPD thickness. Different from the current density–voltage characteristics, the current efficiency–current density characteristics showed an improved efficiency with a 50 nm TPD thickness. The current efficiencies of a device with a 5 nm TPD thickness at a driving voltage of 10 V was 0.148 and at a 50 nm TPD thickness 0.993 cd/A, which was 6.7 times higher than the 5 nm TPD thickness. These results indicated that hole transport in Organic Light-Emitting Diode (OLED) devices were more efficient with thin 5 nm TPD than with thick 50 nm TPD, while electron transport was more efficient with thick 50 nm TPD, which caused conflicting results in the current efficiency-current density and current density-voltage characteristics according to TPD thicknesses.

## 1. Introduction

A display is an image device in which an electrical signal from an electric device is converted to image information so that a person can see the transformed signals as an image on a screen. There are various kinds of displays from the early cathode-ray tubes (CRTs) to liquid crystal displays (LCDs) and organic light-emitting diodes (OLEDs). OLED displays offer a number of advantages—they are self-emitting devices without a backlight as in LCDs, which allows them to be thinner, and they have a fast response time and flexibility [1]. On the other hand, OLEDs present problems that need to be solved such as reducing the power loss of the devices by lowering the driving voltages and improving luminance by increasing emitting efficiency. Among the many attempts to improve these drawbacks, it has been important to optimize the structures of OLED devices. 

Typical OLED device structures consist of an anode for the positive electrode, a hole injection layer, a hole transport layer, an emission layer, an electron transport layer, an electron injection layer, and a cathode for the negative electrode. Indium-tin oxide (ITO) and Ag are the most commonly used anode materials. The ITO properties of low sheet resistance and optically transparent characteristics in the visible wavelength regions make it possible to extract more light from the devices [2]. Silver has a low sheet resistance and a ductile nature [2]. A hole injection layer has been adopted to control the barrier height between the Femi level of ITO and the highest occupied molecular orbital (HOMO) level of the hole transport layer. Using the hole injection layer improves the surface roughness of the ITO. Copper (II) phthalocyanine (CuPc) and poly (3,4-ethylenedioxythiophene) polystyrene sulfonate (PEDOT:PSS) are commonly used for a hole injection layer. In particular, CuPc as a hole injection layer is more thermally and oxidatively stable compared to other organic materials [3]. m-TDATA, molybdenum oxide (MoO_x_) and 2,2′-Dimethyl-*N*,*N*′-di-[(1-naphthyl)-*N*,*N*′-diphenyl]-1,1′-biphenyl-4,4′-diamine (α-NPD) are used as hole transport layer materials [4,5,6]. m-TDATA is based on triphenylamine and has a transition temperature of 75 °C [4]. By using tris-(8-hydroxyquinoline) aluminum (Alq_3_) as a light emitting layer, MoO_x_ can easily transfer charges between the conduction level of MoO_x_ and the HOMO level of Alq_3_ [6,7]. It has been reported that when MoO_x_ is deposited as a thin film, it acts as a charge generation layer, which improves the conductivity of the devices [8]. Mori and Park studied the effects of α-NPD thickness on the electrical conductivity and discussed the electrical conduction mechanism using the space charge-limited current (SCLC) model [6]

Many researchers have studied to improve the performance of OLEDs by lowering the driving voltage of the device and increasing the luminous efficiency [9,10,11,12,13,14,15,16]. In this study, we investigated the effects of the thickness of *N*,*N*′-diphenyl-*N*,*N*′-di(m-tolyl)-benzidine (TPD) as a hole transport material on electrical and optical properties of OLED devices. 

## 2. Experiment 

The ITO substrates were prepared by patterning a product of Samsung Corning (Chungcheongnam-do, Korea) and wet-etching it in an aqua-regia solution. After etching the ITO substrate, it was sufficiently washed with deionized water to remove any residual impurity. We examined the patterned ITO substrate with an optical microscope and measured the sheet resistance.

The ITO substrate cleaning procedure that was performed before depositing the OLED layer was as follows. The substrate was immersed and cleaned ultrasonically in a certain ratio of solution with ammonia (NH_4_OH), distilled water, and hydrogen peroxide (H_2_O_2_) to remove ions, metals, and other contaminants on the surface at 50 °C for 20 min. After it was cleaned, the substrate was dried with a N_2_ gun and then dried on a hot plate at 90 °C for 30 min. A self-assembled monolayer (SAM) from gelest was used as the hole injection material; TPD was used as the hole transport material; and tris(8-quinolinolato) aluminium (Alq3) was used as the emission material.

OLED layers were deposited using a thermal evaporator system. The chamber was evacuated to 1.0 × 10^−5^ Torr before deposition and the deposition rates of organic and metal materials were 0.5–1.0 Å/s and 6.0 Å/s, respectively. A Keithley 617 electrometer (Solon, OH, USA) and a Keithley 236 source-measure unit were used to measure the luminance, current density, and voltage of the OLED devices. The organic layer thicknesses were measured using an Alpha-Step 200 (α-step 200, KLA-Tencor Corporation, Milpitas, CA, USA).

## 3. Results and Discussion

The OLED devices were made of ITO (180 nm)/SAM/TPD (5, 10, 25, 50 nm)/Alq_3_ (60 nm)/Al (100 nm) in which ITO, SAM, TPD, Alq_3_, and Al were the anode, hole-injection layer, hole-transport layer, emitting layer, and cathode, respectively. 

Figure 1 shows the current density-voltage characteristics of OLED devices at various TPD thicknesses. The plot indicates that the current densities increased with decreasing TPD thicknesses at a constant driving voltage. This phenomenon is associated with the relations for the conduction current density in conductors: J=σE, σ=qμn, E=V/d , where J, σ, q, μ, V, and d are the current density, conductivity, charge, mobility, applied voltage, and conductor thickness, respectively. The above equations predict that the current density decreases with an increasing layer thickness. Figure 1 also shows that the current density differences between the various TPD thicknesses increased with an increasing driving voltage. The results indicate that thick TPD layers are not desirable for producing efficient hole-transport layers because of degraded current characteristics per unit area by applying thick TPD layers.

Figure 2 shows the operating voltages of the OLED devices according to the combined thicknesses of an emitting layer of Alq_3_ (60 nm) and a hole-transport layer of TPD (5, 10, 25, 50 nm). As shown in Figure 2, the operating voltages were gradually increasing at the values of 4.0, 6.1, 6.7, and 7.0 V as the combined thicknesses increased at a current density of 1 mA/cm^2^. A similar tendency was observed: Operating voltages increased with increasing combined thicknesses even at higher current densities. This result is also logically expected from the above relations for the conduction current density in the conductor.

Figure 3 shows the effects of the thickness of the organic layer on the current densities. As mentioned above, the thickness of the organic layer was the combined layer thicknesses of an emitting layer of Alq_3_ and a hole-transport layer of TPD. Figure 3 revealed that the current densities were decreasing at values of 9.997, 2.402, 1.344, and 1.011 mA/cm^2^ as the organic layer thickness increased at an operating voltage of 7 V. As the operating voltages decreased to 10 V, the current densities increased drastically at the values of 68.633, 26.673, 16.327, and 7.672 mA/cm^2^ in the same increasing thickness order. We anticipated these current density-organic layer thickness characteristics from the above operating voltage-organic layer thickness characteristics where the operating voltages increased with increasing thicknesses.

Figure 4 shows the luminance characteristics of OLED devices according to the applied current densities. As shown in Figure 4, the luminance efficiency was improved with an increasing TPD thickness, which was in contrast to the current density-voltage characteristics of Figure 1, where the current densities increased with decreasing TPD thicknesses. Thin TPD was more efficient in the hole transport as compared to the electron transport. This kind of unbalanced carrier transports created an inefficient luminance at a thin layer of TPD. On the other hand, there was improved luminance efficiency with a thick TPD layer because of the balanced transports of electrons and holes through the transport layer. This suggested that it is crucial to optimize the carrier transport layer thickness to obtain an efficient luminance and a low operating voltage.

Figure 5 shows the current efficiency-current density characteristics of OLED devices at various TPD thicknesses. Maximum current efficiencies were 0.1, 0.3, 1.1, and 1.5 cd/A with the TPD thickness of 5, 10, 25, 50 nm, respectively. As can be seen in Figure 5, there were noticeably low current efficiencies with 5 and 10 nm TPD thicknesses, while rapidly improving current efficiencies were measured with 25 and 50 nm TPD thicknesses. Figure 6 shows the power efficiency-current density characteristics according to TPD thickness. The plot shows a phenomenon similar to the current efficiency-current density characteristics of Figure 5. The above results confirmed that the electro-optical characteristics of OLED devices with the structures investigated in this research were dominated by hole transports through the hole-transport layer.

## 4. Conclusions

We examined the effects of TPD thickness on the electro-optical characteristics of OLEDs. The current density decreased with an increasing TPD thickness. The results are in agreement with the conduction current density relations with the thickness of the conductor in which the current density is in inverse proportion to the conductor’s thickness. The operating voltages consistently increased as the TPD thicknesses increased at a constant current density. The luminance characteristics of OLED devices according to the current densities improved with an increasing TPD thickness. Poor luminance efficiency with thin 5 nm TPD was due to the insufficient electron supplies through the carrier transport layer, while hole supplies passed relatively easily through the layer. On the other hand, the luminance characteristics improved with the balanced electron and hole supplies through the carrier transport layer with 50 nm TPD. The current efficiency-current density characteristics showed a tendency similar to the luminance-current density characteristics. The maximum current efficiency of the OLED device with a thickness of 50 nm TPD was eight times higher than that with the 5 nm TPD thickness. These results confirmed that optimized control of the hole transport layer thickness was critical to obtaining improved electro-optical properties of OLED devices.

## Figures and Tables

**Figure 1 materials-12-00966-f001:**
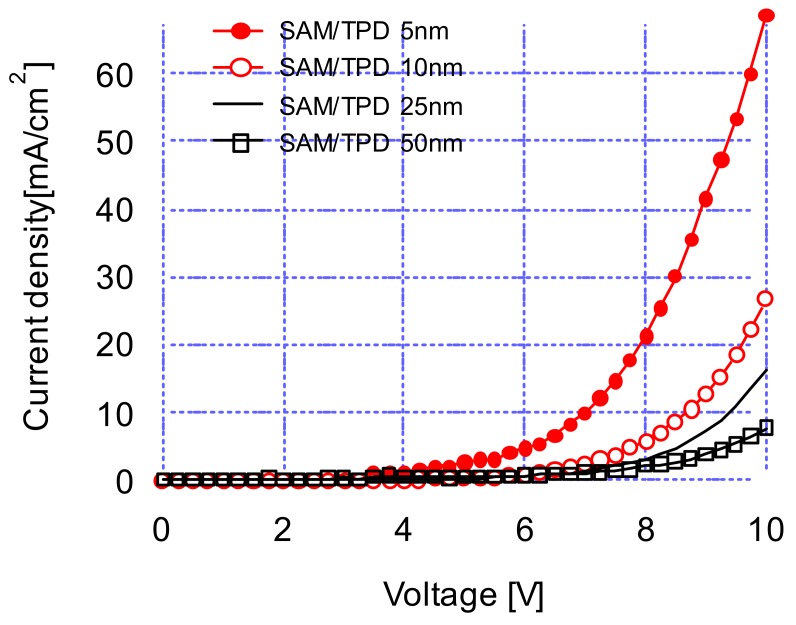
Current density-voltage characteristics of OLED devices at 5, 10, 25, 50 nm TPD thicknesses.

**Figure 2 materials-12-00966-f002:**
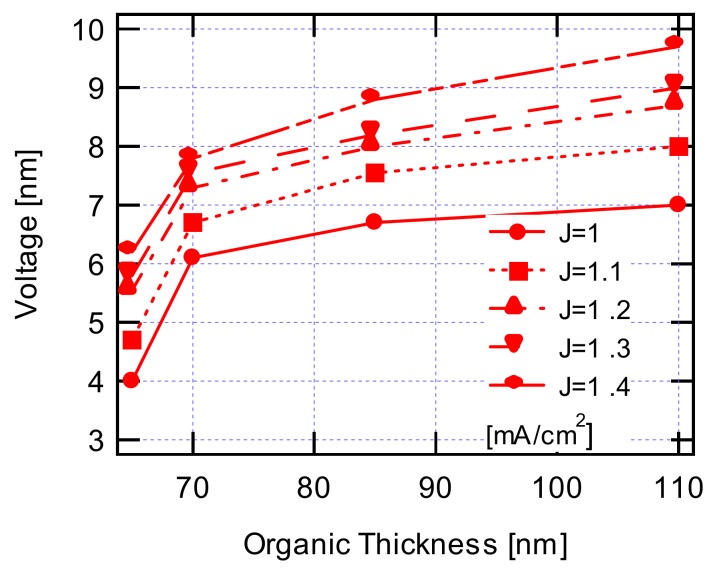
Operating voltage characteristics as a function of organic thickness (emission layer and hole transport layer) at various current densities.

**Figure 3 materials-12-00966-f003:**
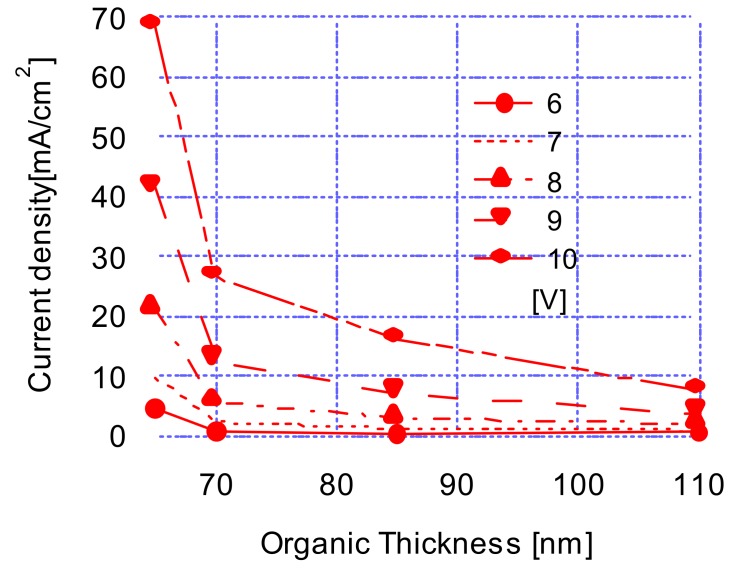
Current density characteristics as a function of organic thickness (emission layer and hole transport layer) at various operating voltages.

**Figure 4 materials-12-00966-f004:**
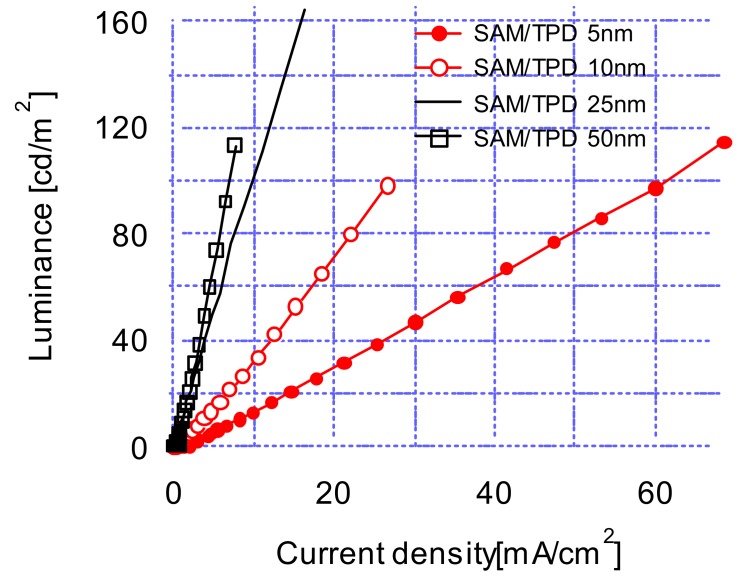
Luminance-current density characteristics of Organic Light-Emitting Diodes (OLED) devices at 5, 10, 25, 50 nm *N*,*N*′-diphenyl-*N*,*N*′-di(m-tolyl)-benzidine (TPD) thicknesses.

**Figure 5 materials-12-00966-f005:**
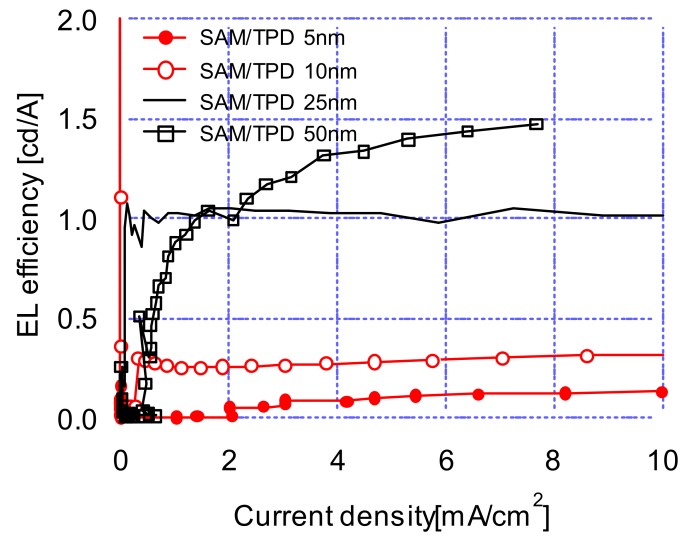
Current efficiency–current density characteristics of OLED devices at 5, 10, 25, 50 nm TPD thicknesses.

**Figure 6 materials-12-00966-f006:**
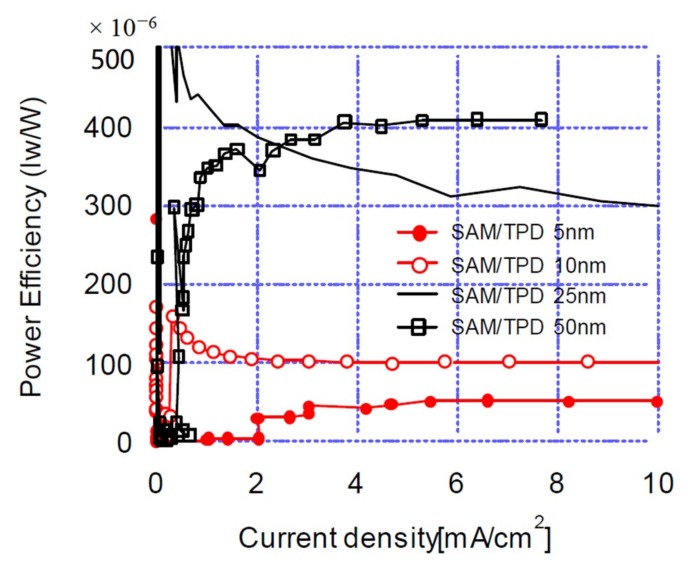
Power efficiency-current density characteristics of OLED devices at 5, 10, 25, 50 nm TPD thicknesses.

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
