# Peer review of "Effects of the Thickness of N,N′-diphenyl-N,N′-di(m-tolyl)-benzidine on the Electro-Optical Characteristics of Organic Light-Emitting Diodes"

_materials, 2019, doi:10.3390/ma12060966_

Reviewer 1 Report

Thanks for offering the great works of your manuscript entitled “Effects of N,N'-diphenyl-N,N'-di(m-tolyl)-benzidine Thickness on the Electro-optical Characteristics of Organic Light Emitting Diode”. Unfortunately, this manuscript is not suitable to publish in Materials in current form. The authors should polish the “introduction” and “experimental results” sections in this manuscript.

 1. This study shows thickness effects on the electro-optical characteristic. However, the introduction does not cite related references about thickness effects and does not illustrate the motivation of using N,N'-diphenyl-N,N'-di(m-tolyl)-benzidine in this study.

 2. In the introduction, the words “CRT”, “LCD”, and “OLED” are abbreviations. These words should show the full names.

 3. Is the thickness related with resistance?

 4. According to the equation in P2L81, the thickness affects the current density. The thickness increases with decreasing current density. However, in P4L106-108, the sentences “As the operating voltages increased to 10 V, the current densities increased drastically at the values of 68.633, 26.673, 16.327, and 7.672 mA/cm2 in the same increasing thickness order” show contrary view to the equation in P2L81.

 5. In the abstract, the authors illustrate “The current efficiencies of a device with 5 nm TPD thickness at the driving voltage of 10 V were measured 0.148 and 0.993 cd/A with a 50 nm TPD thickness, which showed 6.7 times higher at 50 nm TPD thickness.” However, the authors illustrate “Maximum current efficiency of OLED device with 50 nm TPD thickness was 8 times higher than that with 5 nm TPD thickness. These results indicated that the optimized controls of the hole transport layer thickness were critical to obtain an improved electro-optical properties of OLED devices.” in conclusions.

The sentences of abstract “current density: 5 nm is higher than 50 nm” are different to the sentences of conclusions “current density: 50 nm is higher than 5 nm”. Which one is correct? The authors should recheck the experimental results.

Author Response

This study shows thickness effects on the electro-optical characteristic. However, the introduction does not cite related references about thickness effects and does not illustrate the motivation of using N,N'-diphenyl-N,N'-di(m-tolyl)-benzidine in this study.

A : We have added the related references in lines 54-56 from the top of page 2:

   2,2′-Dimethyl-N,N′-di-[(1-naphthyl)-N,N′-diphenyl]-1,1′-biphenyl-4,4′-diamine (α-NPD) are used as a hole transport layer materials [4-6]

   We also added a sentence in lines 60-62 from the top of page 2:

   Mori and Park studied the effects of a-NPD thickness on the electrical conductivity. They discussed the electrical conduction mechanism using the space charge-limited current (SCLC) model [6]

   Motivation

  In this paper, we studied the thickness effects of N,N'-diphenyl-N,N'-di(m-tolyl)-benzidine   

  (TPD) as a hole transport material on the electrical and optical properties of OLED devices  

  in our evaporator system with specified deposition conditions.

  2. In the introduction, the words “CRT”, “LCD”, and “OLED” are abbreviations. These words should show the full names.

A : We added the full names of abbreviations in lines 34-35 from the top of page 1.

 3. Is the thickness related with resistance?

A : TPD itself is a resistive material. Thus, the thickness of TPD has no relation to resistance.

 4. According to the equation in P2L81, the thickness affects the current density. The thickness increases with decreasing current density. However, in P4L106-108, the sentences “As the operating voltages increased to 10 V, the current densities increased drastically at the values of 68.633, 26.673, 16.327, and 7.672 mA/cm2 in the same increasing thickness order” show contrary view to the equation in P2L81.

A : We used the wrong word by mistake. The corrected sentence (in line 120 of page 4) is as  follows:

  As the operating voltages increased to 10 V, the current densities decreased drastically at the values of 68.633, 26.673, 16.327, and 7.672 mA/cm2 in the same increasing thickness order

 5. In the abstract, the authors illustrate “The current efficiencies of a device with 5 nm TPD thickness at the driving voltage of 10 V were measured 0.148 and 0.993 cd/A with a 50 nm TPD thickness, which showed 6.7 times higher at 50 nm TPD thickness.” However, the authors illustrate “Maximum current efficiency of OLED device with 50 nm TPD thickness was 8 times higher than that with 5 nm TPD thickness. These results indicated that the optimized controls of the hole transport layer thickness were critical to obtain an improved electro-optical properties of OLED devices.” in conclusions.

The sentences of abstract “current density: 5 nm is higher than 50 nm” are different to the sentences of conclusions “current density: 50 nm is higher than 5 nm”. Which one is correct? The authors should recheck the experimental results.

A : “The current density” is higher at a 5 nm TPD thickness than at 50 nm as described in lines 16-18 of page 1. On the other hand, the “current efficiency” is as high at 50 nm as compared to 5 nm as described in lines 21-24 of page 1 and 167-168 of page 7.

  Thus, the maximum current efficiency of the OLED was achieved with the 50 nm TPD thickness.

Reviewer 2 Report

      This manuscript examined the electro-optical characteristics of organic light emitting diode depending on TPD thicknesses. These results indicated that hole transports with OLED devices of a thin 5 nm TPD thickness were more efficient than thick 50 nm TPD. Producing high current density by loading less material is very important and this report is recommended to publish in materials after the following major revisions.

Author Response

1. In page no 1, line 34, there should be reference where you mentioned The advantages of  OLED display include self-emitting devices without a backlight unlike a LCD which makes it thinner

A : We added a reference as you recommended.

1. Bernard Geffroy.; Phipippe le Roy.; Christophe Prat, Organic light-emitting diode (OLED) technology: materials, devices and display technologies, Polym. Int. 2006, 55, 572-582.

 2. In page no 1, line 50, Please keep the space between number and degree centigrade. Also, please check with entire manuscript and correct them

A : We revised as you recommended.

 3. In page no 1, line 54, Please correct the word, researches. It should be researchers

A : We changed a passive sentence into an active one (in line 63 of page 2).

  “Many researchers have studied to improve ---- “

 4. I have noted authors mentioned Fig at some places and Figure at some places. Please make sure it is same everywhere.

A : We changed to “Fig.” for every instance.

 5. The biggest drawback I see in this report is no full experimental data for characterization of OLED devices. Please provide the device characterization data using SEM or Microscopy

A : We didn’t take any picture or photo. Unfortunately, we have tried to take pictures but the specimens were contaminated. We will keep your recommendation for the next research.

 6. How did you measure the device thickness? Please provide the data generated to measure thickness of OLED device.

A : The thicknesses of the layers were measured using Alpha-step (a-step 200). We also mentioned the thickness measuring tool in the experimental section in lines 80-81 of page 2.

Round  2

Reviewer 1 Report

I think the authors modified the papers and made it read better. I will recommend publication. 

Reviewer 2 Report

Thank you for the authors to incorporating all my suggestions. 

This report is recommended to publish in materials without any revisions.